# Family support and prayer are invaluable coping strategies for our recovery: Experiences of persons living with cardiovascular diseases

Ivy Selorm Tsedze[1,2], Frank Edwin[2,3], Bennett Owusu[2], Victor Kwasi Dumahasi[4], Nkosi Nkosi Botha[5]*, Nancy Innocentia Ebu Enyan[6]

1 Department of Adult Health, School of Nursing and Midwifery, University of Cape Coast, Cape Coast, Ghana, 2 Cardiothoracic Centre, Ho Teaching Hospital, Ho, Ghana, 3 University of Health and Allied Sciences, Office of the Pro-Vice Chancellor, Ho, Ghana, 4 Institute of Environmental and Sanitation Studies, Environmental Science, College of Basic and Applied Sciences, University of Ghana, Legon, Ghana, 5 Department of Health, Physical Education and Recreation, University of Cape Coast, Cape Coast, Ghana, 6 Department of Public Health Nursing, School of Nursing and Midwifery, University of Cape Coast, Cape Coast, Ghana

* saintbotha2015@gmail.com

**Data Availability Statement:** All relevant data are within the paper.

## Abstract

### Background

It is estimated that 61% of deaths caused by Cardiovascular Diseases (CVDs) globally are attributed to lifestyle-related risk factors including tobacco use, alcohol abuse, poor diet, and inadequate physical activity. Meanwhile, inadequate knowledge and misperceptions about CVDs are disproportionately increasing the prevalence of CVDs in Africa. Moreover, pre-diagnosis awareness/knowledge about CVDs among patients is essential in shaping the extent and scope of education to be provided by healthcare workers.

### Objective

Explore the experiences of patients living with CVDs (PLwCVDs) and accessing care at the Cardiothoracic Centre, Ho Teaching Hospital, regarding their knowledge of CVD-risk factors and coping strategies that work for them.

### Method

We leveraged descriptive phenomenological design to explore the experiences of patients accessing care at the Cardiothoracic Centre at the Ho Teaching Hospital, Ghana. Using the purposive sampling technique, 17 PLwCVDs for 3yrs and above were recruited and included in the study. Thematic analysis was conducted using the qualitative computerised data software, NVivo version 14. Recruitment of participants and general conduct of the study commenced on November 1, 2023 and ended on April 30, 2024.

**Funding:** The author(s) received no specific funding for this work.

**Competing interests:** The authors have declared that no competing interests exist.

**Abbreviations:** CVA, Cerebrovascular accident; DPD, Descriptive phenomenological design; HTH, Ho Teaching Hospital; LMICs, Low- and middle-income countries; NCDs, communicable to non-communicable diseases; PLwCVDs, Persons living with Cardiovascular Diseases; WHO, World Health Organisation.

## Findings

We found that PLwCVDs and accessing care at the Cardiothoracic Centre at Ho Teaching Hospital have adequate pre-diagnosis awareness about CVD-risk factors and their knowledge of same was optimal. Moreover, past unhealthy lifestyles (such as poor dieting, abuse of alcohol, smoking, and inadequate physical activity) may have contributed to participants developing the CVDs. Furthermore, prayers and participation in church activities were the main coping strategies employed by the participants in dealing with CVDs.

## Conclusion

The PLwCVDs and receiving treatment at the Cardiothoracic Centre at the Ho Teaching Hospital are knowledgeable in the CVD-risk factors and adopting positive coping strategies. The Cardiothoracic Centre and government must explore social media facilities to intensify public education and also correct misconceptions about CVDs.

## Introduction

Cardiovascular Diseases (CVDs) have long threatened the psycho-physical and economic health and well-being of humanity, and now counted among the top five causes of mortality and morbidity globally [1–3]. The global burden of CVDs-and-related conditions accelerated in the last 20years, from 271 million cases to 523 million between 1990 and 2019, respectively [4–6]. Moreover, global statistics show that CVDs accounted for nearly 16 million deaths in 2005, over 17.8 million deaths in 2015, and well over 20 million deaths in 2020 [7–9]. The World Health Organisation (WHO) projected that over 23 million people are likely to die from CVDs-and-related conditions by 2030 [10,11]. Meanwhile, 61% of deaths caused by CVDs-and-related conditions globally are attributed to modifiable CVD-risk factors including tobacco use, alcohol abuse, high body mass index (BMI), high blood pressure, high cholesterol, poor dietary choices, high blood glucose levels, and inadequate physical activity [10–12]. Therefore, controlling the modifiable risk factors through lifestyle changes will significantly impact the non-modifiable risk factors and improve the overall health and well-being of, especially, persons diagnosed of these diseases (CVDs-and-related conditions).

So far, the fight against this existential threat to global public health and well-being has been mixed and largely modest [5,13,14]. Meanwhile, global public health efforts have prioritised preventive and promotive strategies aimed at significantly improving public knowledge about the CVD-risk factors, especially, the modifiable risk factors [8,15]. Consistent with this, the Sustainable Development Goal (SDG) 3.4 advocates for healthy lives and the promotion of well-being for all, by significantly reducing the unacceptable loss of lives due to CVDs-and-related conditions [1,10–12]. Thus far, public knowledge about the CVD-risk factors is largely suboptimal [4,5,15], although the situation is reported to be better in the Western world [13,14,16]. Pharmacological and non-pharmacological interventions are needed in the overall management of CVDs-and-related conditions [7,8,17]. However, the evidence [1,11] suggest that improving knowledge about CVD-risk factors among persons living with CVDs (PLwCVDs) is a far more sustainable approach to accelerating early recovery. Moreover, improved knowledge about CVD-risk factors among PLwCVDs will enable them adopt positive lifestyle changes, and also engender better adherence to professional advice from healthcare professionals regarding medications, diet, physical activity, and coping strategies for early

recovery [9,10,17,18]. The argument is that, for persons already diagnosed of and living with CVDs-and-related conditions, improving their knowledge about the CVD-risk factors will make them key actors in their own recovery process.

The African continent is witnessing a record epidemiological transition of diseases from the communicable to non- communicable diseases (NCDs), impacting all groupings with diverse demographics [13,14,16,19–21]. It is estimated that nearly 80% of CVD-burden world-wide and 86% of all cerebrovascular accident cases occur in Low- and Middle-Income Countries (LMICs) [13,14,20]. While multiple factors account for the high rate of CVDs in Africa, inadequate knowledge and misperceptions about the risk factors are disproportionately complicating the situation [22–24]. Moreover, a large proportion of Africa's population is yet to accept the nexus between CVDs and lifestyle [24,25]. The socio-economic burden of CVDs-and-related conditions on Ghana is consistent with the rest of sub-Saharan Africa (SSA), with an increasing number of youths, especially, engaged in lifestyles that predispose them to these diseases [26,27]. For instance, one-fifth of total mortality in Ghana is attributable to CVDs-and-related conditions [28–30]. Also, pre-SARS-CoV-2 evidence [31–34] suggest that from 2010–2019, incidence and mortality due to CVDs-and-related conditions in Ghana remained worryingly high, yet projected to worsen by 2030.

In Ghana, studies on CVDs-and-related conditions exist [26,27,35–44]. Though a reasonable number of studies [31–34,43,45] investigated awareness and knowledge about CVDs in Ghana, there seems to be a gap in literature regarding the experiences of persons diagnosed of and receiving care for CVDs-and-related conditions. Moreover, studies in Ghana that specifically explored the experiences of PLwCVDs concerning their knowledge CVD-risk factors, self-reported risk factors, and coping strategies to CVDs-and-related conditions are lacking in literature.

## Objective of the study

The main objective of this study is to explore the experiences of patients accessing care at the Cardiothoracic Centre, Ho Teaching Hospital, Ghana, regarding their knowledge of CVD-risk factors and coping strategies that work for them.

## Specific objectives

To help achieve the main objective of the study the following specific objectives were applied:

1. To explore awareness and knowledge about the risk factors of cardiovascular diseases among patients accessing the Cardiothoracic Centre at Ho Teaching Hospital.

2. To examine the self-reported risk factors for cardiovascular disease among patients accessing the Cardiothoracic Centre at Ho Teaching Hospital.

3. To examine the coping strategies adopted by patients accessing the Cardiothoracic Centre at Ho Teaching Hospital.

## Materials and methods

### Research design

We leveraged the qualitative research approach, specifically, the descriptive phenomenological design (DPD), to explore the experiences of patients accessing care at the Cardiothoracic Centre at the Ho Teaching Hospital, Ghana [46]. We specifically explored their knowledge, self-reported risk factors, and coping strategies adopted in dealing with the CVDs-and-related

conditions. We considered this design appropriate because it helped to easily explore the lived experiences of PLwCVDs-and-related conditions [47]. Therefore, the phenomenological strategies such as description, variation, bracketing, and introspection were deployed to ensure the findings reflect the subjective experiences of the PLwCVDs-and-related conditions [48,49]. Thus, the variation technique helped unravel various aspects of experiences of the PLwCVDs-and-related conditions [50]. In addition, introspection aided in a deeper understanding of the subjective experiences [51], while bracketing helped mitigate the influence of assumptions, pre-conceptions, and pre-suppositions on the part of the researchers in order to retain the purity of the subject views of the participants [47]. Again, the descriptive strategy helped to deeply immense the researchers into the lived experiences of the PLwCVDs-and-related conditions [47].

Contrary to other forms of design, the DPD is a participant-based approach that upholds the circumstances and context of the participants [46]. Additionally, the DPD can be very useful in unveiling phenomena that eludes quantitative techniques [52–55].

## Study area

The study was conducted at the Ho Teaching Hospital (HTH), in the Volta Regional Capital. The hospital is located off the Ho-Denu road and was established in the year 2000. It is the latest out of the five public teaching hospitals in Ghana, with a bed capacity of 340 and provides specialist services including cardiothoracic services. Though majority of the patients that access the facility are from the host region (Volta Region), Ghana, patients from the neighbouring regions like the Eastern, Oti, and Central regions also access the healthcare at the facility [56]. In addition, the facility also receives patients come from the neighbouring Republic of Togo.

## Sampling technique

The Ho Cardiothoracic Centre has since its operationalisation provided various services to nearly 1600 patients [56]. Most of these patients do not return to the facility after their first visit and also live outside the Volta Region, Ghana. Of this number, there are only 56 PLwCVDs-and-related conditions who live in the Volta Region and regularly access care at the Ho Cardiothoracic Centre [56]. Consistent with the inclusion criteria, we included only patients who have been living with CVDs-and-related conditions for 3yrs and above. We presumed that these patients have the rich experiences that would help address the objective of the study. Therefore, patients who have been living with CVDs-and-related conditions for 3yrs and above, but who were either not available during the period of the study or too ill to actively participate in the study or too emotional in their contributions were excluded from the study. Thus, we leveraged on the purposive sampling technique to recruit and include 17 patients living with CVDs-and-related conditions for 3yrs and above in the study [57–59]. Recruitment of participants and general conduct of the study commenced on November 1, 2023 and ended on April 30, 2024.

## Data collection instruments

Consistent with the objective of the study and broad consultation of relevant literature [27,38,60–62], data were gathered using a semi-structured interview guide. The guide was divided into three sections (A, B, and C) with 5 prompts in all. This included demographic data covering age, gender, level of formal education, duration living with the CVDs, and occupation. Section "AI" includes awareness about CVD-risk factors with 1-prompt and Section "AII" entails knowledge about the CVD-risk factors with 2-prompts. Section "B" covers the

self-reported CVD-risk factors with 1-item, and Section "C" covers the coping strategies adopted with 1-prompt.

Sample items on the interview guide include: *"Tell me a bit about how you first got to know about CVDs"*, *"Tell me what you know about the factors that can positively or negatively affect your condition (CVD)"*, *"Describe the kind of foods that can help control CVDs"*, *"Describe some aspects of your life that you think could be contributing to your condition"*, and *"Describe some of the things you do to enable you cope with this condition (CVD)"*. The study was guided by the following research objectives: i) explore awareness and knowledge about the risk factors of cardiovascular diseases among patients accessing the Cardiothoracic Centre at Ho Teaching Hospital, ii) examine the self-reported risk factors for cardiovascular disease among patients accessing the Cardiothoracic Centre at Ho Teaching Hospital, and iii) examine the coping strategies adopted by patients accessing the Cardiothoracic Centre at Ho Teaching Hospital.

## Pre-testing of instruments

The semi-structured interview guide was pre-tested to ensure it accurately measures the research objective of the study. Two CVD-patients diagnosed at the Volta Regional Hospital were interviewed using the guide. Audio recordings from the pre-testing were transcribed verbatim into typed text while handwritten field notes re-organised [50,63]. In addition, the transcribed text and field notes were reconciled to develop one complete transcript and analysed using the descriptive phenomenological design [64]. Moreover, thematic data analysis was conducted using the qualitative computerised data software, NVivo version 14 [55]. Again, to attain credibility of the instrument, inconsistencies and grammatical errors observed during the pre-testing were resolved. Apart from these changes, we specifically requested for knowledge on the modifiable/behavioural/lifestyle risk factors separately from the non-modifiable/genetical factors. This was not done in the pre-testing. Additionally, we specifically and separately requested to know how modifiable risk factors (excluding diet) and diet had contributed to the patients developing the conditions (CVDs).

Furthermore, the instrument was assessed by author 2 who is Professor in Cardiology, author 3 who is also a seasoned Cardiologist, and 2 Medical Officers at the Medical Out Patients Department of the Volta Regional Hospital, for content validity. All authors went through the instrument again to refine the wording and sharpen the quality of the prompts [50].

## Data collection procedures

Data collection was triggered right after approval of research protocols by the Ho Teaching Hospital Research Ethics Committee (Protocol ID No: HTH-REC [28] FC_2023). Data collection was carried out through face-to-face interviews with the participants at the end of their hospital visit for the day. However, patients who were not willing to participate in the study right after the day's visit were contacted later at home through a special arrangement at their convenience. The interviews lasted between 30–45 minutes per session. The aim and importance of the study were explained to the patients after which written and verbal consents were provided. The aim and study protocols were explained to the 17 patients living with CVDs in English Language and Ewe, local dialects, for ease of comprehension. Fourteen patients gave written informed consent while three gave verbal informed consent. The verbal informed consent process was recorded in the participant's study records or field notes. This included the date, time, and location of the informed consent process. In addition to that, each member of the data collection team that obtained the informed consent wrote the name and signed against the participant's informed consent records. This was approved by the ethical review

board. Moreover, all the 17 patients living with CVDs-and-related conditions for 3yrs and above were assured of their anonymity and the freedom to exit from the study any time without any implications. Again, we conducted the study in cognizance of the Helsinki Ethical Principles for Medical Research involving human subject research. Each participant also signed or gave verbal informed consent before taking part in the study.

## Data processing and analysis

The lived experiences of CVD-patients accessing care at the Ho Cardiothoracic Centre was explored. The DPD was deployed in analysing all data and was done concurrently with data collection [65,66]. Audio recordings from the field were played again and again and manually transcribed verbatim into typed text, while the field notes were carefully re-organised [65]. After which the transcribed data and field notes were reconciled into one complete transcript which was read and re-read severally, together with note making to ensure coherence of thoughts and ideas in the text. Additionally, the audio tapes were played repeatedly and the transcript read over and over again. The essence of this was to get the researchers fully immersed into the data and familiar with the thoughts and ideas articulated by the participants [53]. Data collection and analysis continued until data saturation attained. This stage of the data analysis was conducted by four authors (IST, NIEE, FE, BO, & BNN).

The second step involved reading again and again the full transcript and listening repeatedly to the audio recordings with the view to clearly noting the emerging trends and patterns that evolve into codes [55]. The data coding was conducted by four (NIEE, FE, VKD, & BNN) and verified by two authors (BO & IST). The third step involved a full audit of all the emerging codes to ensure internal homogeneity among similar codes and external heterogeneity among different codes [53]. In addition, each participant was given their transcripts to verify and expatiate on the thoughts and meanings attributed to them. Aspects of the transcripts that the participants disagree with were accordingly revised. Thus, data inconsistencies, repetitions, and omissions were resolved. Moreover, similar codes were grouped together to produce minor themes while the minor themes were further grouped into major themes [48,66,67]. This stage was conducted by four authors (IST, BO, VKD, & BNN) and verified by two authors (NIEE & FE). Finally, the step four involved defining and organising the emerging themes using the qualitative computerised data software, NVivo version 14, and report writing [55,68]. This stage was conducted by three authors (NIEE, FE, & BNN).

## Qualitative rigour

Trustworthiness is regarded a fundamental principle that provides a reliable framework for upholding the constructivists worldview to research [47–51]. Therefore, we adopted five important strategies including credibility, dependability, transferability, reflexivity, and confirmability [47–51]. Thus, credibility was ensured through a careful and persistent observation of the study participants and meticulous and prolonged assessment of data, in search for patterns within and across participants [49,51]. Additionally, member-checking was conducted to avail the participants opportunity to confirm the accuracy of interpretations attributed to them [47,48]. Furthermore, dependability was attained by providing a thorough account of the research approach and design employed and also all decisions leading to the findings of the study. This is to allow for reproducibility by other researchers [47,48,50].

In addition, transferability was ensured through a detailed description of the backgrounds of all study participants and the research approach and design applied in the study to serve as reference for other researchers. The backgrounds of the researchers also helped to enrich and deepen the overall conduct of the study and ensured that the study context was accurately

**Table 1. Summary of themes.**

| Research Objective | Themes |
|---|---|
| Objective One | i) Adequate pre-diagnosis awareness about CVDs. |
| | ii) Good knowledge about CVD-risk factors. |
| | iii) Misconceptions about CVD-risk factors. |
| Objective Two | Unhealthy lifestyles and relationship issues |
| Objective Three | Appropriate coping strategies. |

represented [48,49,51]. However, to ensure that the previous knowledge and preconceptions held by the researchers do not unduly influence the views of the participants, the principle of reflexivity was applied [46,47,51]. Finally, to establish confirmability, techniques such as researcher bracketing was used to mitigate researcher biases [55], while the views of the participants were accurately presented and used to either confirm or discount findings from previous studies [50,53].

# Results

Five themes emerged from the data analysis, that is specific objective one yielded three themes, specific objective two yielded one theme, and specific objective three yielded one theme. Table 1 provides summary of themes that emerged from the analysis while Table 2 provides details of the demographic characteristics of participants.

## Theme one: Adequate pre-diagnosis awareness about CVDs

We found that PLwCVDs-and-related conditions (13 out of 17) had adequate pre-diagnosis awareness about CVDs-and-related conditions. Their awareness about CVDs came mainly through radio, church, television, social media, hospital, workplace, and school. On this issue, a 60yr-old male participant, P4M0620F, remarked:

**Table 2. Demographic characteristics of participants.**

| Variable | | Frequency |
|---|---|---|
| **Gender** | Male | 11 |
| | Female | 6 |
| **Age** | 30–40 | 2 |
| | 41–51 | 2 |
| | 52–62 | 6 |
| | 63–73 | 3 |
| | 74–84 | 4 |
| **Level of formal education** | Middle School | 1 |
| | Secondary/Technical | 1 |
| | Post-Secondary | 9 |
| | Tertiary | 6 |
| **Occupation:** | | |
| Healthcare Worker | | 1 |
| Teacher | | 2 |
| Trader | | 3 |
| Farmer | | 1 |
| Public Servant | | 2 |
| Fireman | | 1 |
| Student | | 1 |
| Retiree | | 5 |

*It was at church that I first heard about this disease (CVD). . .. The church took a decision to organise regular health talks and screening for all members after a member collapsed in church and died two days later. . .. Therefore, it was in 2010, during one of such health talks that I first heard about this disease (CVD).* (Living with CVD for 3years).

Another participant, a 71yr-old female participant, P8F1730RN, commented:

*I studied about non-communicable diseases in school as part of the courses I did. Therefore, in my case, I knew a lot about CVDs for about 46years now. Of course, I also had family members who were diagnosed of CVDs before I did.* (Living with CVD for over 3years).

Another participant, a 76yr-old male participant, P17M6720HI, explained:

*I first heard about this disease (CVD) for over 30yrs from my brother who is a Medical Doctor. Therefore, I can say that I had a good idea about the things (risk factors) that contribute to this disease (CVDs).* (Living with CVD for over 4years).

Moreover, we found that most of the participants were educated by the nurses about the possible causes of their diagnosis (CVDs) and what could be done to guarantee early recovery. Therefore, a 44yr-old male participant, P5M4430PS, remarked:

*. . .in fact, the Doctors and Nurses did well in explaining the factors that may have contributed to my condition (CVD) to me. . . .I now have a better understanding of the factors that could cause this condition (CVD).* (Living with CVD for 3years).

Another participant, a 46yr-old male participant, P9M6440F, said:

*. . .the Doctors and Nurses really spent time in explaining how I may have developed this condition (CVD) to me. This has helped me in avoiding some unhealthy foods and habits that may worsen my condition.* (Living with CVD for 4years).

We were not surprised at the findings given the relatively good educational backgrounds of the participants, especially that some of them were even health professionals or related to health professionals. Fig 1 at the appendix provides details of words used by participants in response to questions on awareness about CVD-risk factors.

## Theme two: Good knowledge about CVD-risk factors

We also found that the PLwCVDs-and-related conditions had good knowledge about CVD-risk factors. This was expected as most of the participants have good educational backgrounds and with few being either health professionals or relations of health professionals. On this issue, participant P4M0620F, remarked:

*I am aware that the foods I eat and the drinks I take together with regular exercise will help improve my condition. But, bad habits like smoking, which I previously do but now stopped, alcohol abuse, high salt and sugar intake can negatively affect my health and may even lead to my death. But, I must confess that it is difficult to stop a habit you are addicted to. In addition to that, if you are a man above 40years, then you can easily develop these diseases (CVDs).*

Another participant, P5M4430PS, explained:

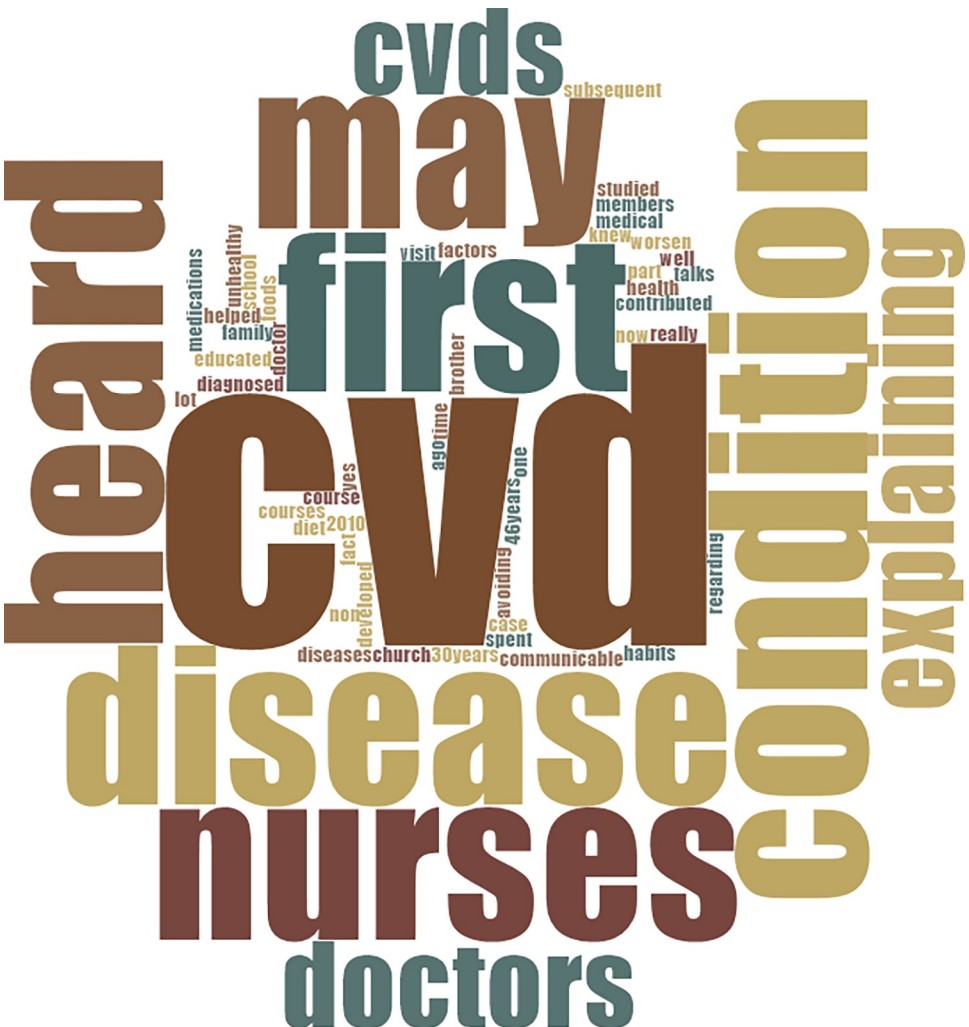

**Fig 1. Words used by participants in response to questions on awareness about CVD-risk factors.**

*. . .oh, this includes what I eat, regular exercise, and certain personal habits. Apart from these, I know that being a man and considering my age and family history, I could be at risk of these diseases (CVD). There is history of these diseases in my family. . ..*

Furthermore, we found that the participants had good knowledgeable about the kinds of foods that could help control conditions (CVDs) and improve their overall health. On this issue, a 76-yr old male participant, P12M6730RT, noted:

*. . .my favourite meals are now boiled yam/cassava/plantain and vegetable stew with fruits, and occasionally fufu and banku with vegetable soup. Some of the vegetables I use include spinach, cucumber, carrot, 'kotomre', cabbage, lettuce, and garden egg. . .oh, the fruits are the ones we have around like orange, mango, pawpaw, coconut, watermelon, and banana. (Living with CVD for 3years)*

Another, a 39yr-old male participant, P14M9320T remarked:

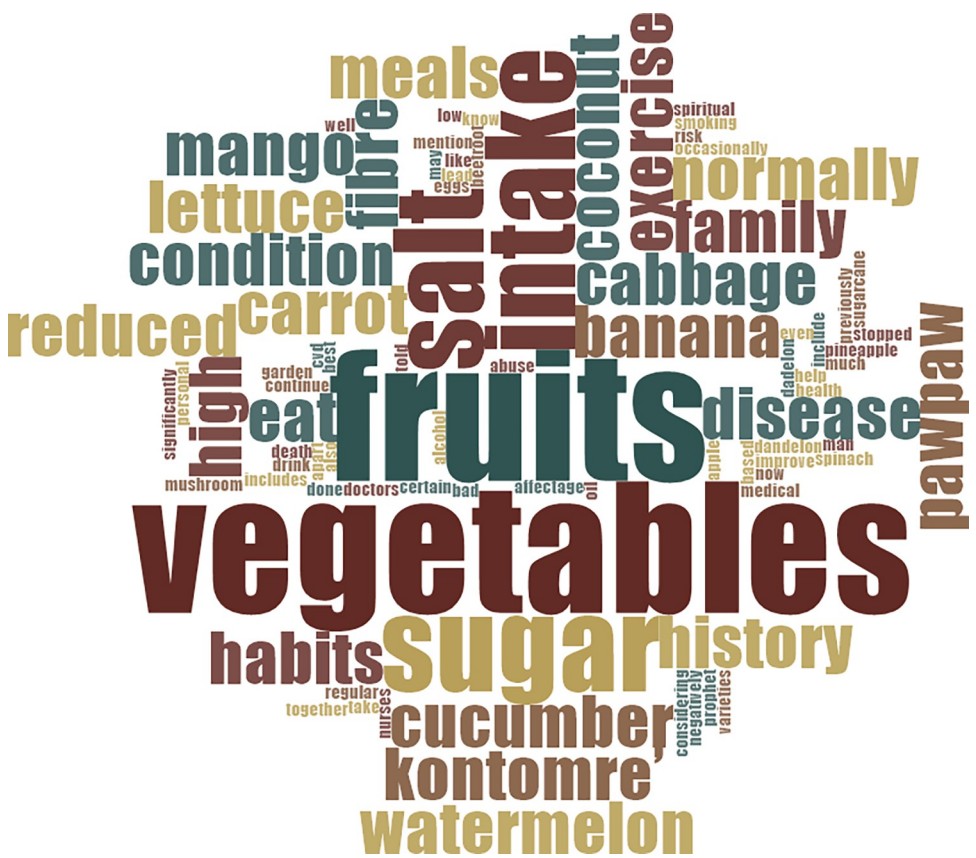

**Fig 2. Words used by participants in response to questions on knowledge about CVD-risk factors.**

*. . .my meals are fibre-based, with varieties of fruits and vegetables, low in salt and oil, while my sugar intake has also reduced so much. As for the vegetables, I can mention 'kontomre', cabbage, carrot, garden eggs, lettuce, cucumber, mushroom, and dadelon. The fruits are normally coconut, watermelon, mango, banana, and pawpaw.* (Living with CVD for 2years). Fig 2 at the appendix provides details of words used by participants in response to questions on knowledge about CVD-risk factors.

**Theme three: Misconceptions about CVDs-and-related conditions.** Interestingly, some participants believed that though the scientifically established CVD-risk factors contributed largely to their conditions (CVDs-and-related conditions), they were convinced that evil forces (spiritual attacks) played a role too. We did not anticipate this, given the relatively good knowledge demonstrated by the participants about the CVD-risk factors and their good educational backgrounds. However, we were equally not surprised at the finding given the overbearing role and influence of religion (spirituality) on the Ghanaian population. On this, a 57-yr old female participant, P13F7570T, remarked:

*. . .but not everything is science. . .I believe what my prophet told me about my condition. He said it was more spiritual than medical. . .. So, apart from the advice from the nurses, I also rely very much on my prophet for direction. I know what I am going through. . ..* (Living with CVD for 7years).

Another, participant P15F8582T, said:

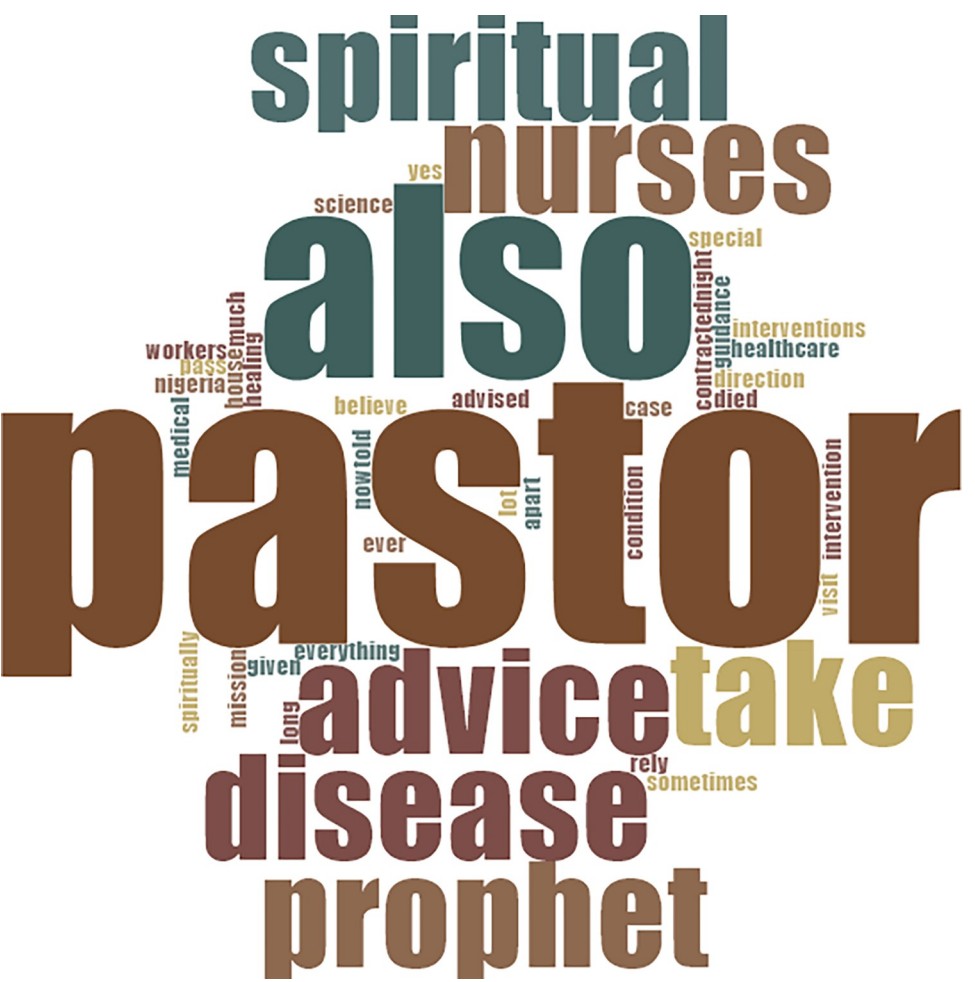

**Fig 3. Words used by participants in response to questions on misconceptions about CVD-risk factors.**

*. . .but for the intervention of my Pastor, I would have died from this disease long before now. I sometimes have to pass the night at the mission house for special spiritual interventions from my Pastor. I would not mention names, but would you believe it when I tell you that a nurse at this hospital suggested I seek "spiritual" support for this condition (CVD)? She advised I visited a Pastor in Nigeria for healing. . ..*

Another, a 58yr-old female participant, P15F8582T, explained:

*. . .my condition was sudden and followed a dispute I had with my uncle over a piece of land in my village. . . .oh yes, my condition was caused by the "evil powers" of that man. . ..* (Living with CVD for 28years). Fig 3 at the appendix provides details of words used by participants in response to questions on misconceptions about CVD-risk factors.

## Theme four: Unhealthy lifestyles and relationship issues

We also found that unhealthy lifestyles previously lived by most of the participants could have contributed to their current conditions (CVDs-and-related conditions). Some of these past

unhealthy lifestyles included poor dieting, alcohol abuse, smoking, and inadequate physical activity. On this issue, a 71-yr old male participant, P1M1753RM, explained:

*You know, I became addicted to cigarette at age 23 and could not stop until I was diagnosed of this disease (CVD). Although I exercised regularly, I could not resist the temptation of smoking. . .I even tried marijuana at a point. Yeah, it was that bad. . .. (Living with CVD for 35years)*

Another, a 68-yr old male participant, P2M8601PS, remarked:

*I sustained a fracture on my left leg since age 14 which affected my level of physical activity. I became overweight as a consequence and could not engage in even simple exercises. My sugar and salt intake were also pretty high. . .. (Living with CVD for 10years)*

Meanwhile, we were pretty surprised that the participants could not leverage their relatively good educational backgrounds and good knowledge about the CVD-risk factors to avoid lifestyles that were injurious to their health.

Furthermore, we found that some participants also had psychosocial issues (such as stress, depression, etc.) that contributed to their conditions (CVDs). Most of these issues were job losses, relationship issues, and other family issues. Thus, a 44yr-old male participant, P5M4430PS, said:

*. . .what I think may have contributed to my conditions (CVDs) is stress. I had issues in my relationship which affected me seriously. It got to a point where I could not sleep well without taking painkillers. . .. However, I can say that I am much better now. (Living with CVD for 3years).*

Another, 52yr-old female participant, P7F2530T, retorted:

*I suffered serious abuses in my two previous relationships which affected my life; I could not concentrate on anything and even ended up at a prayer camp. That was the main cause of my condition (CVDs). . . That experience affected my eating habit, physical activity level, and I became addicted to alcohol. (Living with CVD for 3years).* Fig 4 at the appendix provides details of words used by participants in response to questions on self-reported CVD-risk factors.

### Theme five: Appropriate coping strategies

We also found that the participants adopted appropriate coping strategies in dealing with their conditions (CVDs-and-related conditions). Family support was widely reported as a very helpful coping strategy by the participants. On this, participant P1M1753RM, remarked:

*. . .I cannot do anything on my own without the support of my children and siblings. . .my feeding, going to the hospital for review, bathing, buying of medications, and other daily activities are the responsibility of my children and siblings. I could not have survived without them. . .this is not a condition (CVD) one could deal with all alone.*

Another participant, P2M8601PS, said:

*. . ..my wife and children are the reason I am still alive. They provide me with all the needed support including going to church, making sure I take my medications, preparing my meals,*

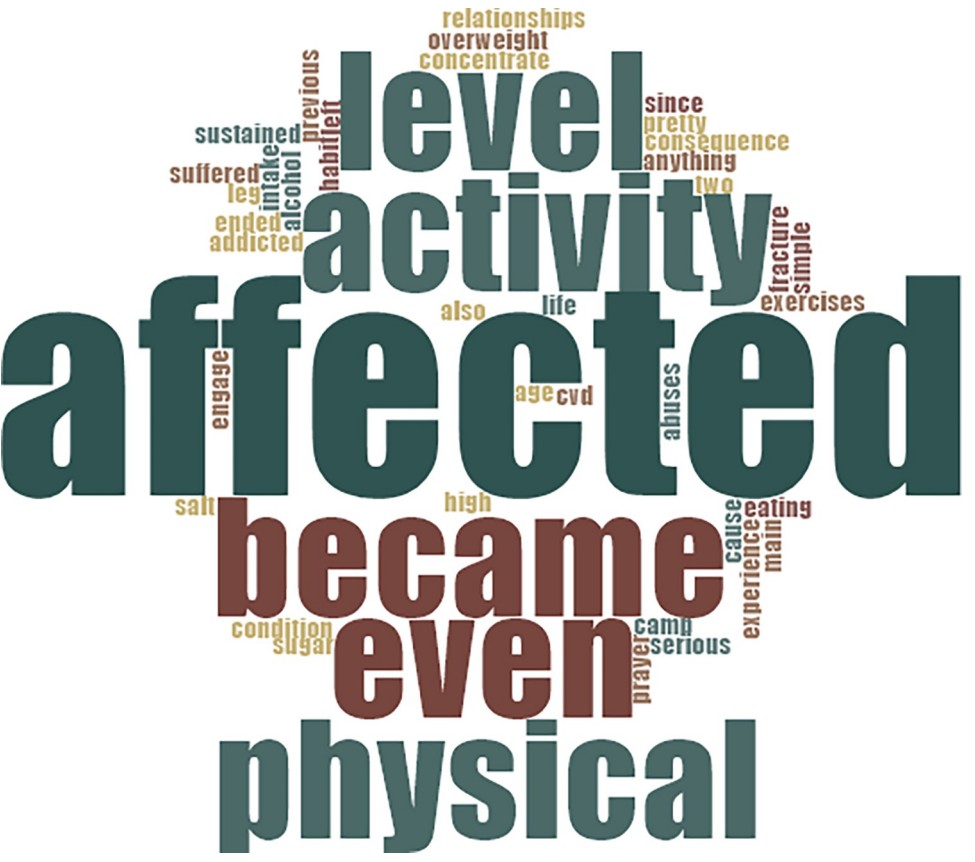

**Fig 4. Words used by participants in response to questions on self-reported CVD-risk factors.**

*and their words of encouragement alone are enough for me. . .they make me think less of the disease.*

Another participant, P3M7770RT, explained:

*. . .as you can see, I am in a wheel chair and cannot do much for myself. Moreover, I am old and do not have the physical strength to deal with this condition (CVD). Thanks to my daughter for being there for me. . .she takes care of all my needs and constantly assures me that I will get better.*

This did not surprise us, as most communities in Ghana still practice the extended family system where members provide support in diverse ways to each other.

Furthermore, we also found that religion (spirituality) was a common coping strategy adopted by the participants in dealing with their conditions (CVDs-and-related conditions). Most of the participants relied very much on prayers and church activities. On this issue, a participant, P5M4430PS, noted:

*A life without Christ is meaningless. . .I am now closer to Christ than before this illness (CVD). As I participate more in church activities, my spirit is encouraged and I have hope that Christ will have mercy on me and grant me full healing from this condition (CVD). Therefore, my weapon against this disease (CVD) is prayer, prayer, prayer. . ..*

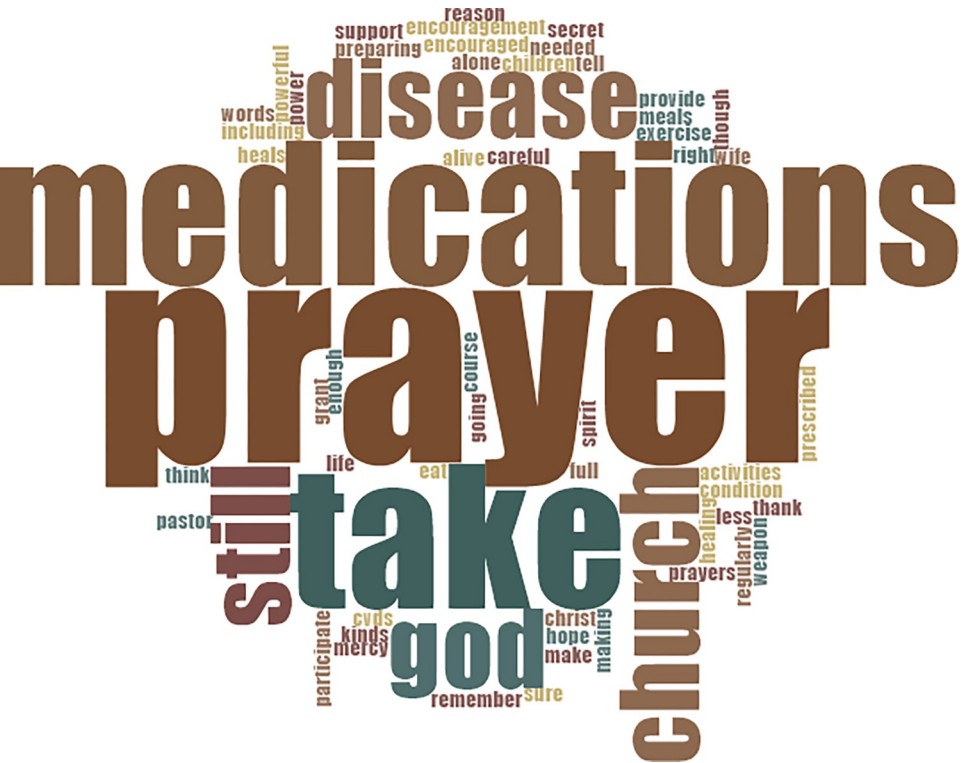

**Fig 5. Words used by participants in response to questions on coping strategies to CVDs.**

Another participant, P7F2530T, explained:

*Just take a good look at me. . .would you believe I had stroke before? That is the power of God for you. . .I thank my Pastor for the powerful prayers over my life. Of course, I still take all the prescribed medications, but remember that it is God that heals. Though I am careful about what I eat, exercise regularly, and take my medications, I can tell you that prayer is my secret.*

We expected this, given that most Ghanaians are religious (spiritual) or believe in some form of deity. Fig 5 at the appendix provides details of words used by participants in response to questions on coping strategies to CVDs.

## Discussion

Overall, the study revealed that PLwCVDs-and-related conditions and receiving treatment at the Cardiothoracic Centre at the Ho Teaching Hospital, Ghana, were adequately aware about CVD-risk factors before being diagnosed of disease, and their knowledge of same was optimal. Additionally, PLwCVDs have good knowledge about the kinds of foods that could help control and improve their conditions (CVDs). Furthermore, past unhealthy lifestyles (such as poor dieting, abuse of alcohol, smoking, and inadequate physical activity) and toxic relationships of the PLwCVDs-and-related conditions may have contributed to them developing the CVDs. However, some PLwCVDs attributed their conditions to evil forces (spiritual attack). More-over, prayers and participation in church activities were the main coping strategies employed by PLwCVDs in dealing with their conditions.

### Adequate pre-diagnosis awareness about CVDs

Pre-diagnosis awareness about CVDs-and-related conditions among patients diagnosed of CVD is essential in shaping the extent and scope of education to be provided by the healthcare workers (HCWs) (Medical Officers and Nurses) [29,30,39,69]. It is much easier and effective for the HCWs to leverage relevant previous information of the patient about CVDs in providing the needed education for early recovery. Several previous studies [22–24] suggested that typically, PLwCVDs-and-related conditions with adequate pre-diagnosis information about CVD-risk factors have better recovery rate than others. Consistent with this, findings of the current study showed that most of the participants have adequate pre-diagnosis awareness about CVDs-and-related conditions.

However, findings of the current study disaffirm findings of previous studies [22–24] that reported poor awareness about CVDs among participants. Meanwhile, it is significant to note that these previous studies were not conducted among PLwCVDs-and-related conditions and receiving treatment. Additionally, findings of the current study align with findings of previous studies [29,30,39,69] that reported that patients diagnosed of CVDs-and-related conditions were educated by HCWs on the possible causes of their diagnosis and lifestyle changes needed to guarantee early recovery. Therefore, it is envisaged that the adequate awareness found in the current study would influence positive behaviour change among the patient.

### Good knowledge about CVD-risk factors

Furthermore, several studies [18,22,29,30] suggested that improving knowledge about CVD-risk factors, especially the modifiable predictors, can significantly impact the recovery rate among PLwCVDs-and-related conditions. Moreover, previous studies [31–34] speculated that improving knowledge about CVD-risk factors can work for Ghana and change the narrative for the better. Therefore, the focus of Ghana's MoH on CVDs-and-related conditions is to accelerate public knowledge about the CVD-risk factors, especially the modifiable factors, as a sustainable way forward on the fight against these diseases [17,70–77]. Consistent with this, findings of the current study showed that the participants had good knowledge about CVD-risk factors. However, findings of the current study contradict findings of several previous studies [36,38,78–82] that reported low knowledge about CVD-risk factors among the respondents.

There are implications of findings of the current study for the participants and public health policy. Given the generally adequate awareness/knowledge recorded among the participants, adherence to instructions on medications and diet maybe reasonably good. Moreover, existing policy strategies towards public awareness/knowledge about CVD-risk factors seem to be working, at least based on the findings of the current study. Therefore, existing strategies may have to be improved to ensure an even better public awareness/knowledge about the CVD-risk factors.

### Misconceptions about CVD-risk factors

Furthermore, a significant drawback to the fight against CVDs-and-related conditions in Africa is public misconceptions about the diseases [8,15]. Previous studies [22–24] a comparatively better public perception about CVDs-and-related conditions than in Africa with high public misconceptions about the disease. For instance, a large proportion of the population in SSA does not know about its hypertensive status and does not seem to realise the correlation between hypertension and CVDs [29,30,69]. Therefore, misconceptions about CVDs-and-related conditions among PLwCVDs-and-related conditions presents a bigger challenge, as that could undermine lifestyle and behaviour change efforts aimed at early recovery [8,23].

Consistent with this, findings of the current study revealed that some participants have misconceptions about CVDs. Most of these participants believed that their conditions (CVDs) came through "spiritual" attacks. It is significant to note that this finding may well reflect the socio-cultural settings of the study area, where believe in the "supernatural powers" is widespread, even among the educated and well exposed [73]. This affirms findings of previous study [13] that suggested that misconceptions about CVD-risk factors conflict with the education provided by the HCWs to the patients.

Findings of the current study have implications for the participants and public health policy. Participants who maybe addicted to their past unhealthy lifestyles may need expert support in adhering to instructions given by the HCWs (Nurses and Doctors). In the absence of this, these participants may relapse and resort to their old bad ways which would undermine recovery. Similarly, the misconceptions about CVD-risk factors may lead to non-adherence to medication and diet instructions given by the HCWs. In terms of public health policy, in addition to existing strategies, public education targeting prevailing misconceptions could be considered.

## Unhealthy lifestyles and relationship issues

The correlation between lifestyle and the development of CVDs-and-related conditions is widely reported [10–14,18,20]. Indeed, previous studies [10–12] reported that 61% of deaths caused by CVDs-and-related conditions globally are attributed to lifestyle (modifiable) CVD-risk factors including tobacco use, alcohol abuse, high body mass index (BMI), high blood pressure, high cholesterol, poor dietary choices, high blood glucose levels, and inadequate physical activity [10–12]. Therefore, controlling the modifiable risk factors through lifestyle changes will significantly impact the non-modifiable risk factors and improve the overall health and well-being of, especially, persons diagnosed of these diseases (CVDs-and-related conditions)

Consistent with these previous studies, findings of the current study revealed that most of the participants developed CVDs as a result of past unhealthy lifestyles. These unhealthy lifestyles included mainly poor dieting, abuse of alcohol, smoking, and inadequate physical activity. This also aligns well with findings of previous studies [13,14,18,20] that reported that lifestyle factors accounted for the highest prevalence of CVDs-and-related conditions in SSA. Moreover, findings of the current study also revealed that some participants may have developed CVDs-and-related conditions as a result of previous psychosocial issues (such as stress, depression, etc.). Most of these were job losses, relationship issues, and other family issues. This relates with previous studies [13,14] that reported similar findings.

## Appropriate coping strategies

Buetow et al. [79] suggested that persons living with chronic conditions would typically adopt many strategies aimed at providing psychological relief from the diseases. Similarly, previous studies [75,76,78–80] explained that, typically, patients confronted with difficult health challenges face three options, including doing nothing about the situation, adopting positive behaviour changes, or adopting negative lifestyles. Thus, whatever the course of action taken, the intention is to ensure that the condition does not emotionally displace the patient [77,81–84]. Consistent with these previous studies, findings of the current study showed that the participants relied on family support in dealing with the demands of their conditions (CVDs-and-related conditions). This upholds previous studies [18,77–84] that reported how family caregivers supported cerebrovascular accident (CVA) survivors in coping with their conditions.

Apart from the physiological support, these family caregivers also provide emotional support to the patients.

Furthermore, findings of the current study revealed the vital role of religion (spirituality) among the coping strategies adopted by the participants. Most of the participants relied very much on prayers and church activities as coping strategies for the CVDs. This coheres with findings of a previous study [73] that reported the role of religion in healthcare. Moreover, a study by Atsu et al. [18] found that family caregivers of CVA survivors relied on prayers as coping strategy for the distressful services they provided.

Findings of the current study do have implications for the health and well-being of the participants and public health policies on CVDs. The two main coping strategies adopted by participants in this study are support from family caregivers and religion. This presents a positive picture of how the education provided by the healthcare workers could be helping the participants in adopting healthy coping strategies. Moreover, policy recommendations on coping strategies may have to be intensified to sustain the gains made so far.

## Contribution to body of knowledge, policy, and practice

Several previous studies [31–34,43,45] have made significant contributions to the subject of knowledge about CVDs-and-related condition in Ghana. However, the current study specifically explored the subject among persons living with CVDs-and-related conditions and the receiving care at the Cardiothoracic Centre. Additionally, the study showed how different populations (educated and non-educated) may differ in their knowledge and coping strategies to CVDs. Thus, health professionals and family caregivers may find the findings useful on how to better provide the needed care to PLwCVDs-and-related conditions. Healthcare managers may also find the study useful in developing policies that would target diverse populations and guarantee better policy outcomes for fighting CVDS. Moreover, to our knowledge, the current study is the first in Ghana to examine pre-diagnosis awareness about CVDs-and-related conditions among PLwCVDs. While demonstrating the novelty of the current study, we are minded that no single study can exhaust all issues regarding knowledge about CVDs-and-related conditions. This informed our suggestions made for future research directions reported in the conclusion.

## Limitations

Notwithstanding its wide application and strength in research, the DPD has some limitations in application [48]. As a key feature, the DPD focuses on the lived accounts of participants and this limits its generalisability of findings [53]. However, this limitation was mitigated through effective researcher bracketing [54]. Again, the DPD is context-specific and therefore limits the generalisability of findings [46]. This was alleviated through a rigorous application of the procedures and proper member-checking [47]. In addition, the potential for researcher-bias is higher in DPD and thus confounds the reliability of the findings [53]. Nevertheless, effective application of bracketing helped to mitigate this [55]. Moreover, the study excluded patients who have been living with the condition (CVDs) for less than 3years. This may result in the loss of information that may otherwise be vital to the study and further limiting the generalizability of the findings of the study. Additionally, the study was limited to the Ho Teaching Hospital among only 17 participants could also limit the generalizability of the findings to other Cardiothoracic Centres in Ghana. Furthermore, though the findings from the study were expected, it may be difficult to firmly establish causality and make inferences due to the approaches employed. A quantitative approach may have clearly established causality and improve the reliability of the evidence.

## Conclusions

The study revealed that PLwCVDs-and-related conditions and receiving treatment at the Cardiothoracic Centre at the Ho Teaching Hospital, Ghana, are knowledgeable in the CVD-risk factors and adopting positive coping strategies in dealing with the conditions (CVDs). Specifically, persons living with CVDs and accessing care at the Cardiothoracic Centre at Ho Teaching Hospital have adequate pre-diagnosis awareness about CVD-risk factors and their knowledge of same was optimal. Moreover, participants know the kinds of foods that could help control and improve their conditions (CVDs). Moreover, past unhealthy lifestyles (such as poor dieting, abuse of alcohol, smoking, and inadequate physical activity) and toxic relationships may have contributed to participants developing the CVDs. However, some participants attributed their conditions (CVDs-and-related conditions) to evil forces (spiritual attack). Furthermore, prayers and participation in church activities were the main coping strategies employed by the participants in dealing with CVDs-and-related conditions.

Consistent with the findings, it is recommended that the Cardiothoracic Centre at Ho Teaching Hospital and the government must, in addition to the traditional mediums of communication, explore social media facilities, and work closely with faith-based centres and faith leaders to promote CVD risk awareness and psychosocial support during treatment. Moreover, future research direction may leverage quantitative approaches to investigate the psychosocial safety hazards among healthcare workers at cardiothoracic centres in Ghana.

## Supporting information

**S1 Checklist. Completed COREQ checklist.**
(DOC)

## Author Contributions

**Conceptualization:** Ivy Selorm Tsedze, Nkosi Nkosi Botha, Nancy Innocentia Ebu Enyan.

**Data curation:** Ivy Selorm Tsedze, Frank Edwin, Bennett Owusu, Victor Kwasi Dumahasi, Nkosi Nkosi Botha, Nancy Innocentia Ebu Enyan.

**Formal analysis:** Nkosi Nkosi Botha.

**Investigation:** Ivy Selorm Tsedze, Bennett Owusu, Victor Kwasi Dumahasi, Nkosi Nkosi Botha, Nancy Innocentia Ebu Enyan.

**Methodology:** Ivy Selorm Tsedze, Frank Edwin, Bennett Owusu, Nkosi Nkosi Botha, Nancy Innocentia Ebu Enyan.

**Resources:** Ivy Selorm Tsedze, Bennett Owusu, Nkosi Nkosi Botha.

**Software:** Nkosi Nkosi Botha.

**Supervision:** Frank Edwin, Victor Kwasi Dumahasi, Nancy Innocentia Ebu Enyan.

**Validation:** Nkosi Nkosi Botha.

**Writing – original draft:** Ivy Selorm Tsedze, Frank Edwin, Bennett Owusu, Victor Kwasi Dumahasi, Nkosi Nkosi Botha, Nancy Innocentia Ebu Enyan.

**Writing – review & editing:** Ivy Selorm Tsedze, Frank Edwin, Bennett Owusu, Victor Kwasi Dumahasi, Nkosi Nkosi Botha, Nancy Innocentia Ebu Enyan.

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
