## [Decision Letter · Decision Letter 0]

26 Nov 2024

PONE-D-24-37089Family support and prayer are invaluable coping strategies for our recovery: Experiences of persons living with cardiovascular diseasesPLOS ONE

Dear Dr. Botha,

Thank you for submitting your manuscript to PLOS ONE. After careful consideration, we feel that it has merit but does not fully meet PLOS ONE’s publication criteria as it currently stands. Therefore, we invite you to submit a revised version of the manuscript that addresses the points raised during the review process.

We look forward to receiving your revised manuscript.

Kind regards,

Engelbert A. Nonterah, MD, PhD

Academic Editor

PLOS ONE

4. Please include your tables as part of your main manuscript and remove the individual files. Please note that supplementary tables (should remain/ be uploaded) as separate "supporting information" files

Reviewers' comments:

Reviewer's Responses to Questions

**Comments to the Author**

1. Is the manuscript technically sound, and do the data support the conclusions?

Reviewer #1: Partly

Reviewer #2: Yes

2. Has the statistical analysis been performed appropriately and rigorously? 

Reviewer #1: N/A

Reviewer #2: N/A

3. Have the authors made all data underlying the findings in their manuscript fully available?

Reviewer #1: No

Reviewer #2: Yes

4. Is the manuscript presented in an intelligible fashion and written in standard English?

Reviewer #1: Yes

Reviewer #2: Yes

5. Review Comments to the Author

Reviewer #1: This is a well written manuscript and timely given the increasing prevalence of CVD in Ghana as well as other parts of the world.

See below specific comments:

Methodology:

- Sample size: can the authors clarify their sample size? Do they mean that only 56 people living with CVDs were attending the cardiothoracic centre at the time of the study? or is the 56 a fixed figure i.e., the maximum capacity of the centre.

- Can the authors be specific about the changes made after pretesting the semi-structured interview guide?

- Did the authors obtain written or verbal consent or both?

- Can the authors provide more details on the positionality of the interviewer and their qualitative research experience? Were the interviews conducted in English or the local language?

- The limitations of the DPD and how they were mitigated can be moved to the limitation section

- Can the authors list the steps involved in their thematic analysis and reference the thematic analysis approach used.

- I strongly recommend that the authors use the COREQ checklist to frame the information presented in the methodology. This would help fill in the gaps and strengthen the study methodology.

Discussion:

- I suggest that the authors include a paragraph providing an overview of all the study findings in relation to the current evidence before delving into discussion of the separate themes.

-Limitations: the authors should further reflect on the limitations of their study and the possible influence of the population selected. For example it is expected and not surprising for people living with CVDs to have better knowledge of risk factors.

- What are the implications of the study findings? What do they mean for future research, policy and practice? This needs to be strengthened.

Conclusion:

- The conclusion does not align with the findings presented in the paper, e.g., family support and prayer. The authors concluding statement emphasises the role of social media, what about working closely with faith-based centres and faith leaders to promote CVD risk awareness and psychosocial support during treatment?

Data availability:

- Please review the data availability statement. The statement "All data underpinning the study are fully captured in the manuscript" is not deemed adequate as the data availability statement is referring to the transcripts/complete data and not the extracts included in the manuscript.

General:

- Review the manuscript thoroughly for grammar and correct use of tenses

Reviewer #2: In addition, the participants were given opportunity to verify the transcript and expatiate on the thoughts and meanings attributed to them. Thus, data inconsistencies, repetitions, and omissions were resolved.

There is no information concerning how the transcripts were given to the participants to verify the transcripts.

There is no information about who did the data coding

No information about the data collectors

6. PLOS authors have the option to publish the peer review history of their article (what does this mean?). If published, this will include your full peer review and any attached files.

Reviewer #1: No

Reviewer #2: No

---

## [Author Response · Author response to Decision Letter 0]

4 Dec 2024

December 4, 2024

POINT-BY-POINT RESPONSE TO REVIEW COMMENTS 

We submit for your consideration point-by-point response to review comments on “Family support and prayer are invaluable coping strategies for our recovery: Experiences of persons living with cardiovascular diseases”. We appreciate the time and effort of the reviewers in going through the manuscript and making such meaningful remarks. Most especially, we consider the remarks very useful in improving the quality of the paper for global audience. Included in this new submission are a marked-up copy of manuscript labelled “Revised Manuscript with Track Changes” and an unmarked version of manuscript labelled “Manuscript”. 

Our specific responses to the review comments are reported below, please. 

We very much appreciate the efforts and time of the editor for the academic observations made which have actually helped in improving the quality of the paper. Our specific responses are as follow. 

Requirement 1: Please ensure that your manuscript meets PLOS ONE's style requirements, including those for file naming. 

Response: Thank you. We have done our best to format the manuscript to conform to PLOS ONE's author guideline.

Requirement 2: Please provide additional details regarding participant consent. In the ethics statement in the Methods and online submission information, please ensure that you have specified (1) whether consent was informed and (2) what type you obtained (for instance, written or verbal, and if verbal, how it was documented and witnessed). 

Response: Thank you. We have provided further and better details regarding participant consent. In the ethics statement in the Methods and online submission information, we specified (1) whether consent was informed and (2) what type we obtained (for instance, written or verbal, and if verbal, how it was documented and witnessed). Please, refer to the track changes copy for details on the revision.

Requirement 3: Your ethics statement should only appear in the Methods section of your manuscript. If your ethics statement is written in any section besides the Methods, please delete it from any other section.

Response: Thank you. The ethics statement appeared in the Methods section only of the manuscript. Please, refer to the track changes copy for details on the revision.

Requirement 4: Please include your tables as part of your main manuscript and remove the individual files. Please note that supplementary tables (should remain/ be uploaded) as separate "supporting information" files

Response: Thank you. We have included the tables as part of the main manuscript and removed the individual files from the system. Please, refer to the track changes copy for details on the revision.

Reviewer 1 

We cannot thank the reviewer enough for these splendid comments made which have helped in improving the quality of the paper. Our specific responses to the suggestions are as follows, please. 

Comment 1: This is a well written manuscript and timely given the increasing prevalence of CVD in Ghana as well as other parts of the world.

Response: We are very honoured to have you review our paper. 

Comment 2: Sample size: can the authors clarify their sample size? Do they mean that only 56 people living with CVDs were attending the cardiothoracic centre at the time of the study? or is the 56 a fixed figure i.e., the maximum capacity of the centre.

Response: Thank you. We have revised the sub-section to ensure clarity on the sampling technique. Please, refer to the track changes copy for details on the revision. 

Comment 3: Can the authors be specific about the changes made after pretesting the semi-structured interview guide?

Response: Thank you. We have revised the sub-section to specifically indicate what changes we made to the instrument following the pre-testing. Please, refer to the track changes copy for details on the revision.

Comment 4: Did the authors obtain written or verbal consent or both?

Response: We have provided further and better details regarding participant consent. In the ethics statement in the Methods and online submission information, we specified (1) whether consent was informed and (2) what type we obtained (for instance, written or verbal, and if verbal, how it was documented and witnessed). Please, refer to the track changes copy for details on the revision.

Comment 5: Can the authors provide more details on the positionality of the interviewer and their qualitative research experience? Were the interviews conducted in English or the local language?

Response: Thank you. We have provided details on the positionality of the interviewers and their qualitative research experience in the competed COREQ checklist. Please, refer to the attached copy for details. On the issue of the language used for the interview, we have provided further details under the sub-section, “Data collection procedures”. Please, refer to the track changes copy for details. 

Comment 6: The limitations of the DPD and how they were mitigated can be moved to the limitation section.

Response: Thank you. We have moved the limitations of the DPD and how they were mitigated the limitations section, as suggested. Please, refer to the track changes copy for details. 

Comment 7: Can the authors list the steps involved in their thematic analysis and reference the thematic analysis approach used.

Response: Thank you. Data processing and analysis were conducted through a 4-step descriptive phenomenological design (DPD). That is, step one involved data organisation and familiarisation, step two involved data synthesis and developing relevant codes, step three involved developing and review of relevant themes, and step four involved defining and organising themes using the qualitative computerised data software, NVivo version 14. Therefore, data processing and analysis were conducted through a 4-step “descriptive phenomenological design (DPD)” and not “thematic analysis” as captured under the data analysis in the manuscript. The error has accordingly been corrected in-text. Please, refer to the track changes copy for details. 

Comment 8: I strongly recommend that the authors use the COREQ checklist to frame the information presented in the methodology. This would help fill in the gaps and strengthen the study methodology.

Response: Thank you. Completed copy of COREQ checklist submitted.

Comment 9: I suggest that the authors include a paragraph providing an overview of all the study findings in relation to the current evidence before delving into discussion of the separate themes.

Response: Thank you. A paragraph providing an overview of all the study findings in relation to the current evidence provided as suggested. Please, refer to the track changes copy for details.

Comment 10: Reflect on the limitations of their study and the possible influence of the population selected. For example it is expected and not surprising for people living with CVDs to have better knowledge of risk factors.

Response: Thank you. The limitations of the study was accordingly revised to reflect the suggestions made. Please, refer to the track changes copy for details. 

Comment 11: What are the implications of the study findings? What do they mean for future research, policy and practice? This needs to be strengthened.

Response: Thank you. The contribution to body of knowledge, policy, and practice was accordingly revised to reflect the suggestions made. Please, refer to the track changes copy for details.

Comment 12: The conclusion does not align with the findings presented in the paper, e.g., family support and prayer. The authors concluding statement emphasises the role of social media, what about working closely with faith-based centres and faith leaders to promote CVD risk awareness and psychosocial support during treatment?

Response: Thank you. The section has been revised to ensure it embraces and aligns with all key findings of the study. Additionally, the concluding statement was revised to capture collaboration with faith-based centres and faith leaders to promote CVD risk awareness and psychosocial support during treatment. Please, refer to the track changes copy for details.

Comment 13: Please review the data availability statement. The statement "All data underpinning the study are fully captured in the manuscript" is not deemed adequate as the data availability statement is referring to the transcripts/complete data and not the extracts included in the manuscript.

Response: Thank you for this suggestion. However, the raw transcripts you request for contained participant information that is deemed confidential and cannot be disclosed to third parties. We wish to reiterate that every data required for the purpose of this articles is dully presented in text, tables, and figures, please. 

Comment 14: Review the manuscript thoroughly for grammar and correct use of tenses

Response: Thank you. The manuscript is thoroughly revised for grammar and correct use of tenses.

Reviewer 2

We cannot thank the reviewer enough for these great comments. We believe these will help in improving the quality of the paper. Our specific responses are as follows. 

Comment 1: In addition, the participants were given opportunity to verify the transcript and expatiate on the thoughts and meanings attributed to them. Thus, data inconsistencies, repetitions, and omissions were resolved. There is no information concerning how the transcripts were given to the participants to verify the transcripts.

Response: Thank you. The sub-section has been revised to reflect the suggestions made. Please, refer to the track changes copy for details.

Comment 2: There is no information about who did the data coding.

Response: Thank you. We have provided details on the authors who conducted the data coding. Please, refer to the track changes copy for details.

Comment 3: No information about the data collectors

Response: We have provided details on the authors who conducted the data collection. Please, refer to the track changes copy for details. 

Yours sincerely,

… SIGNED.…

BOTHA NKOSI NKOSI 

Corresponding Author

---

## [Editor Report · Decision Letter 1]

26 Dec 2024

Family support and prayer are invaluable coping strategies for our recovery: Experiences of persons living with cardiovascular diseases

PONE-D-24-37089R1

Dear Dr. Nkosi Nkosi Botha,

We’re pleased to inform you that your manuscript has been judged scientifically suitable for publication and will be formally accepted for publication once it meets all outstanding technical requirements.

Kind regards,

Engelbert Adamwaba Nonterah, MD, PhD

Academic Editor

PLOS ONE
---

## [Editor Report · Acceptance letter]

10 Jan 2025

PONE-D-24-37089R1 

PLOS ONE

Dear Dr. Botha, 

I'm pleased to inform you that your manuscript has been deemed suitable for publication in PLOS ONE. Congratulations! Your manuscript is now being handed over to our production team.

Kind regards, 

on behalf of

Dr. Engelbert Adamwaba Nonterah 

Academic Editor

PLOS ONE